# Influence of Fermentation Beetroot Juice Process on the Physico-Chemical Properties of Spray Dried Powder

**DOI:** 10.3390/molecules27031008

**Published:** 2022-02-02

**Authors:** Emilia Janiszewska-Turak, Maciej Walczak, Katarzyna Rybak, Katarzyna Pobiega, Małgorzata Gniewosz, Łukasz Woźniak, Dorota Witrowa-Rajchert

**Affiliations:** 1Department of Food Engineering and Process Management, Institute of Food Sciences, Warsaw University of Life Sciences—SGGW, 02-787 Warsaw, Poland; s197299@sggw.edu.pl (M.W.); katrazyna_rybak@sggw.edu.pl (K.R.); dorota_witrowa_rajchert@sggw.edu.pl (D.W.-R.); 2Department of Food Biotechnology and Microbiology, Institute of Food Sciences, Warsaw University of Life Sciences—SGGW, 02-787 Warsaw, Poland; katarzyna_pobiega@sggw.edu.pl (K.P.); malgorzata_gniewosz@sggw.edu.pl (M.G.); 3Department of Fruit and Vegetable Product Technology, Institute of Agricultural and Food Biotechnology, 36 Rakowiecka Street, 02-532 Warsaw, Poland; lukasz.wozniak@ibprs.pl

**Keywords:** color, hygroscopic properties, TGA, FTIR, betalain, chromatography, SEM

## Abstract

Picking vegetables is, along with salting and drying, one of the oldest ways to preserve food in the world. This is the process of decomposition of simple sugars into lactic acid with the participation of lactic bacteria. The aim of the study was to obtain powders from fermented red beet juice with the highest possible amount of lactic acid bacteria (LAB) and active ingredients. For the analysis, juices were squeezed from the vegetables and two types of fermentation were used: a spontaneous fermentation and a dedicated one. After inoculation, samples were taken for analysis on a daily basis. Extract, pH, total acidity, pigments, and color were measured. In addition, microbiological tests were also carried out. The juices from the fifth day of fermentation was also spray dried, to obtain fermented beetroot powder. Juices from 3–5th day were characterized by a high content of LAB and betanin, had also a low pH, which proves that the lactic fermentation is working properly. The exception was the juice from spontaneous fermentation. According to the observations, the fermentation process did not run properly, and further analysis is needed. The powders were stable; however, results obtained from the pigment content and the LAB content are not satisfactory and require further analysis.

## 1. Introduction

Recently, fermented products, including vegetable and vegetable juices, have become more and more popular. This is related to new nutritional trends and growing consumer awareness of food processing [1,2,3].

The main vegetables used in the production of fermented by lactic acid bacteria in Poland are cucumbers, cabbage, and beets. In other countries, vegetables such as broccoli, cauliflower, tomatoes, garlic, onions, white radish, peppers, and olives are also pickled [1,4].

Beetroot and its juice has a number of health-promoting properties. Consuming it helps prevent atherosclerosis, lower cholesterol, increase immunity, reduce stress, and slow down the aging processes. This vegetable, as well as its juice, contains many minerals such as iron, magnesium, calcium, zinc, copper, and manganese. Moreover, beetroots contain a significant amount of folate (90–95 μg/100 g of raw product), carbohydrates, and fiber (1.8 g/100 g of raw product) and are a source of vitamin C [2,5,6]. Red beet also contains biologically active substances, which include betalains, phenols, and B vitamins. The health-promoting properties of the beetroot juices resulting from the content of these ingredients also include: antioxidant, antidepressant, antibacterial, antifungal, anti-inflammatory, diuretic, expectorant, and laxative effects [6].

The silage includes probiotic bacteria, i.e., bacteria that have a beneficial effect on the host’s health by changing the host’s microflora. Only a product containing live microbial cells can be a probiotic, which, after colonization in the human body, will contribute to the improvement of human health with their presence [7,8,9]. Fruit and vegetable products, such as fermented vegetable juices or pickled vegetables and fruits, are an excellent source of probiotics. Additionally, their advantage is the fact that they can be consumed by people with lactose intolerance for whom dairy products are unsuitable [7,8,9].

An important aspect of natural fermentation of vegetables is to maintain the appropriate process conditions. For this purpose, the addition of salt is used, temperature is controlled, and the time of the process needs to be investigated. Due to the use of appropriate salinity, it is possible to maintain the safety and quality of the fermentation process. A salt concentration of 0.5–1% has a positive effect on the growth of LAB bacteria, especially the *Lb. plantarum* species [10]. Microorganisms are a key element in maintaining the efficiency of the ensilage process, so the temperature cannot be too high for them [11]. Temperature affects bacteria in several ways: by regulating their growth rate, enzyme activity, nutritional requirements, and the chemical composition of cells, indirectly influencing the solubility of molecules and compounds present in their cells, regulating diffusion of chemical compounds and ion transport, and changing the osmotic properties of membranes. Therefore, the highest efficiency and biomass production of biomass in the lactic fermentation process while maintaining a temperature in the range of 26–42 °C [1,12]. Furthermore, length of the fermentation process is also important; too long or too short can result in significant decreases in bacteria. The selection of appropriate time is crucial for further use of fermented juice.

To accelerate the fermentation process, starter cultures are added to the silages. They are a selected group of microorganisms that prevent the development of undesirable microflora and increase the efficiency of the fermentation process. Among bacteria, the most commonly used starters are *Lactiplantibacillus plantarum*, *Lactiplantibacillus pentosus*, *Levilactobacillus brevis*, and *Limosilactobacillus fermentum*, while common yeast species are *Wickerhamomyces anomalus*, *Saccharomyces cerevisiae*, *Saccharomyces boulardii*, and *Candida boidinii* [13,14,15,16].

Fermented juices can be treated as a source of the lactic acid bacteria as well as the source of active ingredients, in beetroot juices pigment–betalain. However, the use of liquid form can be difficult in industry processes. That is the reason for the change of the product state from liquid to solid. Spray drying with carrier addition is one of the processes which can be used for creating powder products with a high amount of LAB and pigment content. In this powdered form it can be used as an additive substance to food production [17,18].

In the spray drying process of juices, the addition of a carrier material is crucial. Without a high molecular substance added to the juice, it is almost impossible to spray-dry it, due to the glass transition temperatures, mainly of sugars [18].

To the best of our knowledge, there is no complex information on the process of pickling beet juice with the use of dedicated lactic acid bacteria and then drying it by spray drying. Therefore, the aim of the study was to obtain powders from fermented red beet juice with the highest possible amount of lactic acid bacteria and active ingredients.

The work was divided into two stages: the first stage covered the execution of a spontaneous lactic acid fermentation process and also dedicated fermentation with us of selected starter cultures to determining the highest number of lactic acid bacteria during fermentation; the second step involved the spray drying process of the fermented beetroot juice. At each stage of the experiment, apart from the content of bacteria, the physicochemical factors such as color or pigment content in the juice and powder were assessed. Furthermore, to verify the fermentation process, the pH value and acidity were controlled. In powders, parameters related to subsequent storage, such as structure, degradation temperatures (TGA test), were also determined.

## 2. Results and Discussion

### 2.1. Analysis of the Physicochemical Properties of Fermented Beetrrot Juices

The fermentation process depends on many factors; one is the type of lactic acid bacteria used and the other is the duration of the process. For plant materials, the duration of the process is longer due to the poor availability of sugars and other substances necessary for the development of LAB [19,20]. In the case of vegetable juices, according to the literature, the duration of this process can be shortened. The first part of the research was to determine the most favorable stopping time for the fermentation process of beetroot juice. For this purpose, from the third day after the beginning of the process, characteristic parameters such as pH and acidity, as well as the content of microorganisms and active ingredients (betalain), were determined. Data on this analysis are summarized in Table 1, Figure 1 and Figure 2.

After three days of fermentation, the kinetics of the process was analyzed. In the juices inoculated with the starter cultures, a decrease in pH and an increase in acidity were observed; in the case of spontaneously fermented juice, the pH remained unchanged until the sixth day in relation to the starting juice (Table 1). This may be due to the lower number of bacteria at the beginning of the process of adding juices with the addition of starter culture or inappropriate conditions for this kind of fermentation. The addition of starter cultures is considered to be a factor that significantly accelerates the development of the microflora responsible for the lactic acid fermentation process. The same level of pH was mentioned by [9] for fermented Chokeberry juice as well as for beetroot juice fermented by *Lb. acidophilus LA-5*, or *Lb. rhamnosus* [5].

The extract values and the dry matter content remained unchanged, regardless of the bacteria responsible for the lactic acid fermentation process and the day of the process (Table 1). The extract is related to the soluble solid content in the samples and includes glucose, sucrose fructose, and water-soluble proteins, as well as carbohydrates, organic acids, proteins, fats, and minerals of the water-soluble components of the juice [21].

The color of the sample, according to the literature, is correlated with the pigment content [22,23]. In the case of red beet pigments are betalains, which consist of yellow-orange betaxanthins and red-purple betacyanins. Betalain are pigments that are insensitive to pH changes from 3 to 7, which is the reason of its stability during the fermentation process in which pH decreases for beetroot till 3.5 at maximum decrease [3,19,24]. In juices the highest color coefficients L* were observed in fresh beetroot juice; during the fermentation process, the lightness independently of the type of bacteria used for fermentation. Additionally, a decrease was observed during the time of fermentation. Different situations were observed for other color coefficients; coefficient a*, which represents red-green color, and coefficient b*, related to the yellow-blue color, both increased at the end of the fermentation process. However, for those coefficients (a*, b*), the situation was dynamic, on the third day an increase was observed for the b* coefficient, while for coefficient a* decrease was observed (except values for *Lb*. *plantarum*). In our opinion, differences can be caused only by the type of bacteria used in the process.

The comparison of juices pictures showed that the fermentation process, independently of its type, caused changes in the color of the juices. The fermented juices obtained a shade similar to the purple, compared to beetroot juice, the color of which is similar to brown (Appendix A). No differences between fermentation types was seen, which is the reason of showing only pictures from SF and LP fermentation process.

However, as seen in Table 1, there was no clear correlation in the color changes depending on the day and/or the lactic acid bacteria used. That which was observed as one color for beetroot juices via spontaneous fermentation was very different from other ones (Table 1 and Appendix A).

Analysis of pigment values showed that at the sixth or seventh day of fermentation the obtained values were higher than in previous days. However, this trend has no reflection in statistical analyses, no statistically significant changes in pigment content during fermentation were observed (Figure 1). Czyżowska et al. [25] presented similar values for the fermentation process of beetroot juice for red and yellow pigments. In their research, fermented beetroot juice was pressed and those samples were used for storage. They have obtained results at the level of 69 mg/L for red pigment and 19 mg/L for yellow once. During storage, these values decreased [25].

The analysis of bacterial counts is an important parameter in the evaluation of the fermentation process. Analysis of lactic acid bacteria content on seven days of fermentation process showed that, independently of starter cultures, the stationary phase of growth lactic acid bacteria was observed on the third to fifth day of the process, during this time, the highest number of lactic acid bacteria during fermentation was determined. Previously, pre-tests showed that this phase was from the third to the sixth day (Appendix A). For the spray drying of lactic acid, the stationary phase of growth is recommended due to the stability of its metabolites; cell components are recommended [26]. Lian et al. [27] reported that stage of growth affects the heat resistance of microorganisms, which are least sensitive to heat at their stationary phase [27,28]. Therefore, spray drying of the fermented juices was performed on the fifth day when the cells were in stationary growth phase.

An increase in the number of bacteria by 2–2.5 log CFU/g was observed during the 3 days of fermentation, followed by stabilization at the level of 7.0–7.5 log CFU/g, after the fifth day of fermentation, a decrease in the number of bacteria by 1–2 log cycles was observed, which may be caused by the use of sugars in fermented juices by bacteria. Spray drying reduced the survival rate of lactic acid bacteria from the silage. A 1-log cycle decrease in bacterial viability was observed in the case of spontaneous fermentation and 2–2.5 log cycles in the fermentation using the reference strains. This could be due to improper spray drying parameters for fermented juices or the concentration or type of carrier used.

Spray drying has been used for the production of starter cultures and dehydrated lactic acid bacteria, as the powder obtained can be transported at a low cost and can be stored in a stable form for prolonged periods [29]. The cell viability of individual bacterium may be affected by various factors during the spray-drying process, which occur over a very brief period of time, which makes it difficult to predict the full damage that bacteria cultures incur throughout the drying process [30]. In addition, parameters such as air flow, mechanical processing, storage conditions, and encapsulated material can influence the survivability of bacteria [31]. Presented in this article data for spray drying process were used in accordance our previous experiments for non-fermented beetroot juices [17] and juices obtained from fermented slices of beetroot [1]. Further research requires the refinement of the spray drying method of fermented juices in order to increase the survival rate of lactic acid bacteria.

### 2.2. Analysis of Physicochemical Properties of Fermentated Beetrrot Powders

Day five was chosen for spray drying, based on data from preliminary studies. In preliminary studies, the stationary phase for fermented beetroot juice lasted from day fourth to day sixth and in those days was the largest number of lactic acid bacteria. Therefore, in the presented study, the juice of day fifth was selected as the day when the bacterial count was stable enough for drying the juice by spray drying.

To each juice, 10% *v*/*v* of maltodextrin DE = 10 was added. The samples were stirred until the whole maltodextrin solubility was reached and then transferred to the spray dryer.

Beetroot powders obtained after spray drying fermented juices were characterized by high dry matter content (90–95%), low water activity (for almost all samples) (0.35), high total acidity (TA), approximately 3 g lactic acid/100 g of product (in comparison to the juices where TA was below 1.6 g lactic acid/100 g of product), and low sorption properties 16–20% (Table 2). The kinetics of the synthesis is presented in Appendix A.

The water vapor adsorption kinetics of the food powders tested at water activity 1 showed a tendency to resemble the water vapor adsorption isotherm of the same materials (Appendix A). In the case of the fermented powder, there was a constant increase in humidity, regardless of the bacteria used for fermentation, within 24 h of sorption. Szulc and Lenart [32] showed similar relationships of sorption kinetics for baby or strawberry powder.

The high dry matter and low water activity, as well as the low sorption properties, are a guarantee for the future storage stability of powders. The discussed properties of powders may prevent the development of microorganisms and slow down the processes of chemical transformations that could take place under unfavorable storage conditions [33,34].

Powders obtained after spontaneous fermentation had the lowest dry matter content and the highest water activity, which was also related to the higher sorption of water from the environment. Furthermore those powder were obtained from juice with higher pH values, which drastically differ from the other used for spray drying. Obtained powder was lighter than juices taken to the spray drying; this could be related to the use of maltodextrin addition, which is almost white powder itself [35,36], what is also seen in Appendix A. Juices with added 10% of maltodextrin are lighter than the same juices without it.

The pictures presented in Appendix A are in the same relation as the results obtained by the color analysis (Table 2). The color of the SF_powder differs the most from the other ones, however this could be related to the moisture content of this powder, which was the highest. The higher moisture content can influence the ‘eye’ reflection of samples; due to this it looks lighter.

At the same time, larger amounts of adsorbed water for the SF_powder and LB_powder powders may be related to the larger particles obtained by drying (seen in Appendix A). The particle size could be related to the acidity of the environment (in both cases, on the fifth day it was lower than for LP_5 or LF_5 juices). The high acidity of the juice may cause its slow drying compared to juices with low acidity.

The morphology of the powders obtained did not differ from that obtained from non-fermented beetroot juices [37]. All powders were spherical with dents and wrinkles; damage particles were seen (Figure 3). Similar observations were made for fermented noni-juice [35] and fermented beetroot juices [1]. The main factor determining such a shape of the obtained powder is the spray-drying process itself, using the spray disk as the atomizing mechanism [38].

In the article, two different methods of pigment content determination were used. A spectrophotometric method was used to estimate the total content of betalain compounds (betanin and vulgaxanthin) during the fermentation process. To determine the profile of individual betalain compounds, HPLC analysis was used for the juices from the fifth day; therefore, this was used for spray drying and for the obtained powders. As a result of this analysis, it was possible to present changes in the content of individual compounds after the drying process. Results from HPLC method presented identification of the types of betanin and betaxanthin, as well as the amount of them, are presented in Table 3.

The main component of betaxanthin was stated to be vulgaxanthin I (A column in Table 3) and was detected in all samples. In a sample from fresh beetroot juice was also detected other betaxanthin in a range from 3.5 to 6 mg/100 g d.m. (Table 3). It could be related to the fermentation process and lowering the pH of the juice, while betaxanthin are more sensitive for pH changes than betacyanins [22,39,40]. From betacyanins, the main part of the samples was detected as betanin (G) and isobetanin (I). The identification of other ones showed that a possible decarboxylated derivative of betanin (E), betanidin (J), and neobetanin (M) were observed. In powders, other unidentified chemical compounds from betacyanins were also observed, especially in samples after spontaneous fermentation (Table 3).

The same betacyanins were detected in juices obtained from fermented beetroot by pressing by Wilkowska et al. [41] and Czyżowska et al. [42]. However, the proportion of compounds was different. In their research, the highest amount was observed for isobetanidin, then betanidin, and a similar level was seen for betanin and isobetanin [41]. In our research, the highest amount was seen for betanin, then isobetanidin, and a small amount of decarboxylated derivative of betanin, neobetanin, and betanidin (Table 3).

High degradation of both betacyanins and betaxanthins from fresh beetroot juice during fermentation could be related to the peroxidase enzyme, which can be activated after slicing and pressing juice from the vegetable, which was confirmed by Czyżowska et al. [42]. During fermentation, enzymes can react in juices and cause degradation of betalains. Furthermore, the reason for the huge degradation of betaxanthins from 183 (fresh beetroot juice) to 1.4–0.1 mg/100 g d.m. (Table 3) could be an effect of the temperature of the fermentation process used in this research (26 °C). It is seen in literature that, at this temperature, betacyanins are more stable than betaxanthins [41,42].

The degradation of both betacyanins and betaxanthins after spray drying is related to the process temperature higher than 70 °C. It is related to the exit temperature of the drying air in which the particle stays a few microseconds, which may influence the degradation of the beads. However, as is proven (cited), the addition of a carrier results in greater protection against degradation, which is observed in the studies provided [1,17,18].

In powders, higher amounts of neobetanin (M) and betanidin were seen. Neobetanin is usually formed by cleavage of two hydrogens (formation of a double bond) with betanin or isobetanin; the bond is in such a place that both compounds yield the same product; while betanidin (J) is a compound obtained from betanin after glucose breakdown [43].

#### Thermal Properties of Powders

Thermal analysis such as TGA allow us to determine changes in the state of the tested substance with temperature under the conditions of a given measurement. These methods are used to study chemical reactions and phase transitions that occur during the heating of substances. Moreover, thermal analysis allows us to determine the thermal durability of materials. Thermogravimetry records changes in the mass of a substance during heating or cooling as a function of time and temperature [44].

FTIR spectral analysis of fermented beet juice powder was performed to identify the presence of functional groups (Figure 4), which can be used to confirm compounds such as the presence of betalians.

In analysis of TGA, curves can be steps or region extracted. The first stage between 50 and 120 °C is related to the moisture from material loss; the second step, above 120 °C, matches to the decomposition processes of particle components such as proteins and carbohydrates [45,46,47]. In analyses samples, the highest mass loss (56–58%) was observed in second region (step 2) (Table 4 and Appendix A) which could be related to the maltodextrin present in all samples. The decomposition temperature at the 160 °C level of 160 °C in second region is related to the glass transition temperature for maltodextrin with a low dextrose equivalent [48]. The highest water evaporation at the first step for SF_powders is related to the highest water content in that sample (Table 2 and Table 4, Appendix A).

In the range of 3250–3050 cm^−1^, the single bond area was seen which belongs to hydrogen bond. Peaks in the range 1620, 1144, and 800–600 cm^−1^ showed the presence of hydroxyl compounds. A small peak around 1650 cm^−1^ belongs to carbonyl double bond which can derived from aldehydes, esters, or carboxyl. In fingerprint region (600–1500 cm^−1^) it was seen at 1314, 1350—methyne C-H bend. In all samples, a peak at 1150 cm^−1^ is related to C-O stretching, suggesting the presence of phenols. In turn, peaks between 1415 and 1650 cm^−1^ may indicate the presence of nitrogen-containing functional groups. Betalains are nitrogen-containing compounds in their structure; therefore, these peaks confirm their presence in the sample. FT-IR spectra confirmed differences in the molecular structure of the powders, which proves the change of bioactive compounds in the material. Similar relationships were found for dried beet pomaces [49], beetroot [50,51]. FT-IR spectra have confirmed the presence, in all powder samples of nitrogen-containing compounds, of active ingredients from beetroot.

## 3. Materials and Methods

### 3.1. Materials

Beetroot (Beta vulgaris) was purchased from a local supermarket (Warsaw, Poland) and stored in a temperature range 4–6 °C maximum for 2 days before used. As an inoculum for fermentation Levilactobacillus brevis KKP 804 (LB), Lactiplantibacillus plantarum ATCC 4080 (LP), and Limosilactobacillus fermentum KKP 811 (LF) were used. Reference strains were obtained from the American Type Culture Collection (ATCC, Manassas, VA, USA) and the Collection of Industrial Microorganisms (KKP, Warsaw, Poland).

### 3.2. Technological Treatment

#### 3.2.1. Juice Pressing

The vegetables were used to obtain juices. The process was obtained with a juicer model NS-621CES (Kuvings, Daegu, Korea). Then, the juice was filtered via paper filters to clarify.

#### 3.2.2. Fermentation Process

The fermentation process was carried out according to the Janiszewska-Turak et al. [1] protocol with some changes. Juices were placed in a 200 mL jars. 1% *v*/*v* of NaCl was added. Then, inoculum in the amount of 1% of the juice volume, which corresponded to the bacterial content of approximately 1 × 10^7^ CFU/mL, was added. For anaerobic conditions, jars with juice were closed and kept in an incubator at a stable 26 °C. The fermentation process took 7 days from inoculum addition. From the third day of fermentation, daily analysis was performed. For each day of experiments, separate jars were opened for tests. All experiments were performed in duplicate. The experiments were carried out in parallel.

#### 3.2.3. Spray Drying

Fermented juices with added 10% of carrier (maltodextrin with dextrose equivalent DE = 10, Peepes S.A., Łomża, Poland) were dried in the spray drier LAB S1 (Anhydro, Copenhagen, Denmark) equipped with a spray disc (diameter 0.064 m). Based on previous experiments with beetroot juices [1], the drying process parameters were: spray disc speed of 39,000 rpm; raw material flux rate of 8.3·10^−7^ m^3^/s; inlet air temperature 160 °C; air flow 0.055 m^3^/s; and outlet air temperature 70 ± 2 °C. Powder was collected in the cyclone and then transferred to glass ‘twist-off’ jars. Dried powders were stored in jars in a dark place. The experiments were performed in duplicate.

### 3.3. Analytical Method

#### 3.3.1. Dry Matter

For dry matter determination, the gravimetric method was used. About 0.6–1 g of juice was placed in a dish with filter paper; for powders, 1 g was placed into a dish without filter paper, and dried by vacuum drying method (Memmert VO400, Schwabach, Germany) under the pressure of 10 mPa in 75 °C for 24 h until constant weight, according to information from [2,52]. Measurements were made in triplicate.

#### 3.3.2. Soluble Solid Content in Juices

Measurement of soluble solid content in juices was performed with a Pocket Refractometer PAL-3 (ATAGO Instruments, Tokyo, Japan). All measurements were made in triplicate.

#### 3.3.3. pH and Total Acidity

The pH was measured with a pH-meter (S-210, Mettler Toledo, Greifensee, Switzerland) in duplicate for each sample. For total acidity measurement, a known weight of beet juice (*m*) was taken and diluted with distilled water to a volume of 50 mL (*V*_1_), then 25 mL of diluted juice (*V*_2_) was taken for testing. After placing the magnetic stirrer and electrode in the test solution, titration with NaOH solution was started until pH was 8.1, and the amount of NaOH solution used (*V**_NaOH_*) was recorded for calculations. This measurement was performed in duplicate for each sample.
(1)Total acidity=(VNAOH×0.1×V1×0.09×100)(V2×m) [g lactic acid/100 g product]
where 0.1 is the molarity of NaOH and 0.09 is the index for correction for lactic acid in sample

#### 3.3.4. Water Vapor Adsorption

The kinetics of water vapor adsorption was determined using a stand ensuring continuous measurement of changes in mass of the samples. A Mettler AE 240 balance, adapted to continuous operation in conditions of constant temperature and relative air humidity, was used for the tests. The kinetics of water vapor adsorption were carried out in the water activity of the environment (aw = 1) at the temperature of 25 °C for 24 h. The sample for kinetic tests was about 0.5 g of the powder. After placing the sample in the hygrostat, the change in its mass was recorded with the help of the POMIAR computer program. Measurements were made in 2 repetitions.

#### 3.3.5. Color Parameters

Analysis of powder color was made in CR-5 (Konica Minolta Sensing Inc., Osaka, Japan) in CIE L*a*b* system. The measurement parameters are as follows. Illuminant D65, angle 2 ° and calibration with white color as presented by [17]. All measurements were made in 5 repetitions. Furthermore, pictures of juices from fifth day and powders were taken with the methodology of Rybak et al. [53]. The photos were made with a Nikon D7000 digital camera (Nikon, Tokyo, Japan) with a 105 mm lens positioned vertically in front of on a sample (juice) or vertically of the sample (powder) at a distance of 50 cm; the light source consisted of four fluorescent lamps reduced at an angle of 45, emitting daylight at a temperature of 6500 K. Photos were saved in JPG format.

#### 3.3.6. Morphology of Powder Particles

In order to test the morphology of microcapsules, images on a desktop scanning electron microscope, Hitachi TM3000 (Hitachi High-Technologies Corporation, Tokyo, Japan) operated at an accelerating voltage of 10 kV were taken. Powders were kept on the double sticky film placed on the microscope table then covered with gold. Images were taken by a built-in light color optical navigation camera.

#### 3.3.7. Thermal Properties

Thermogravimetric analyses were performed using a TGA thermal analyzer (TGA/DSC 3+, Mettler Toledo, Greifensee, Switzerland) to determine thermal stability and degradation of the powders. The powder sample (approximately 7–8 mg) was heated at 5 °C per min from 30 to 600 °C under nitrogen atmosphere (N2 flow was 50 mL per min). TGA and DTG curves were acquired from the differential TGA values. Two measurements of each sample were made.

#### 3.3.8. FT-IR Spectroscopic Analysis

The spectra of spray dried juices were collected using Cary 630 (Agilent Technologies Inc., Santa Clara, CA, USA) with ATR system. Approximately 2 mg of sample was placed on the diamond crystal. Spectra were recorded in the wavenumber range 4000–650 cm^−1^, with a resolution of 4 cm^−1^, verging 32 scans. Background measurement was performed prior to each sample. Three measurements of each sample were made.

#### 3.3.9. Betalain Content

In the article, a spectrophotometric method was used to estimate the total content of betalain compounds (betanin and vulgaxanthin I) during the fermentation process. To determine the profile of individual betalain compounds, HPLC analysis was made for juices used for spray drying and for the obtained powders.

(A)Spectrophotometric method

To calculate the amount of betalain, first the spectrophotometric method described by Janiszewska-Turak et al. [2] was used. The measurement was carried out at Evolution 220 (Evolution 220, Thermo Scientific, Waltham, MA, USA). Pigments were extracted from the sample with a phosphate buffer at pH 6.5. The 0.5 g of sample (juice or powder) were mixed with 5000 mg of phosphate buffer. The determination of red and yellow pigments was calculated in terms of betanin (mg betanin/ 100 g dm) and vulgaxanthin-I (mg vulgaxanthin-I/100 g dm), respectively. Pigment content calculations were based on the absorption values A1%, which were 1120 for betanin (at 538 nm), 750 for vulgaxanthin-I (at 476 nm), and 600 nm, which is related to the correction of the impurities amounts [2]. Two measurements of each sample were made.

(B)Chromatography method

Sample preparation

Dried samples were grounded to a fine dust with a mortar and a pestle. The portions of approx. 1 g were extracted with 20 mL of 0.2% formic acid and acetonitrile (4:1, *v*/*v*) mixture and shaken for 5 min. Subsequently, the samples were centrifuged for 5 min at 5000× *g* and the supernatants were filtered through 0.45 μm syringe PTFE filters (Macharey-Nagel, Duren, Germany) to chromatographic vials.

Liquid samples were analyzed directly after centrifugation and filtration.

Quantitative analysis of pigments

The analyses were conducted according to a method presented by Ravichandran et al. [54] using an HPLC set: 2695 Separations Module, 2995 Photodiode Array Detector, and 2475 Multi-Wavelength Fluorescence Detector (all Waters, Milford, MA, USA). The separation of 10 µL samples was performed on Sunfire C8, 5 μm, 4.6 × 250 mm^2^ column (Waters) with an appropriate precolumn within 60 min at a column temperature of 30 °C and a flow rate of 1.0 mL/min using gradient of 0.2% formic acid (A) and acetonitrile (B) as follows: 0 min, 100% A; 7 min, 100% A; 10 min, 97% A; 27 min, 90% A; 35 min, 90% A; 45 min, 80% A; 50 min 0% A; 55 min, 100% A; and 60 min, 100% A. The pigments were quantified using spectrophotometric data. For betacyanins, a wavelength of 538 nm was used and the results were expressed as equivalents of betanin, while for betaxanthins a wavelength of 480 nm was used and the results were expressed as equivalents of vulgaxanthin I. Two independent analyses were performed for each sample.

Identification of pigments

The pigments were identified on the results of our previous experiments [55], their UV spectra, and the typical retention order during RP-HPLC separation [56,57,58]. Additionally, betacyanins and betaxanthins were distinguished on the basis of their fluorescence at the excitation wavelength of 465 nm and the emission wavelength of 510 nm. In the betaxanthins, a strong fluorescence occurs due to the presence of four conjugated double bonds, while coupling them to the aromatic ring of cyclo-DOPA in the betacyanins leads to loss of fluorescence [59].

### 3.4. Determination of the Number of Lactic Acid Bacteria

Total count by pour plate method was used to enumerate viable cells. Fermented juice samples were serially diluted using sterile saline (0.85% NaCl, Biomaxima, Lublin, Poland). The samples were streaked onto plates de Man Rogosa and Sharpe Agar (MRS, Biomaxima, Lublin, Poland) agar and incubation at 30 ± 1 °C for 48 ± 4 h. The number of colonies grown was counted (ProtoCOL 3—Automatic colony counting and zone measuring, Synbiosis, Frederick, MD, USA) and recorded as log CFU per g d.m. The samples were analyzed in triplicate.

### 3.5. Statistical Treatment

The results obtained were subjected to a statistical analysis using Statistica 13 software (StatSoft, Warsaw, Poland), using one-way analysis of variance with the Tukey HSD test at a significance level of α = 0.05. The other parameters were determined using MS Excel 16.

## 4. Conclusions

The analysis of the results showed that the fermentation process of beetroot juice from third to fifth day keeps the bacterial count at the same level which allows the use of juices from these days for the spray drying process.

The juices of those days were characterized by a high content of LAB and betalain, and also had a low pH, which proves that the lactic fermentation was working properly. The exception was the juice from spontaneous fermentation. The fermentation process did not run properly in this case (high pH, low LAB content)—a decrease in bacteria content was observed from the 3rd day of fermentation. Further analysis of this type of juice fermentation is needed.

We can conclude that the powders obtained from the fifth day of fermentation were stable, as evidenced by the high content of dry substance, low water activity, and low water adsorption capacity. However, the obtained results of pigment content and lactic acid bacteria content are not satisfactory and require further analysis. FT-IR spectra confirmed differences in the molecular structure of the powders, which proves the change of bioactive compounds in the material, which was also visible in HPLC results. Thermal analysis (FT-IR) could be used for identification of powder compounds. Thermogravimetric analysis (TGA) showed that the powders obtained are stable at ‘room temperatures’; the first degradation temperature started from 58–60 °C.

## Figures and Tables

**Figure 1 molecules-27-01008-f001:**
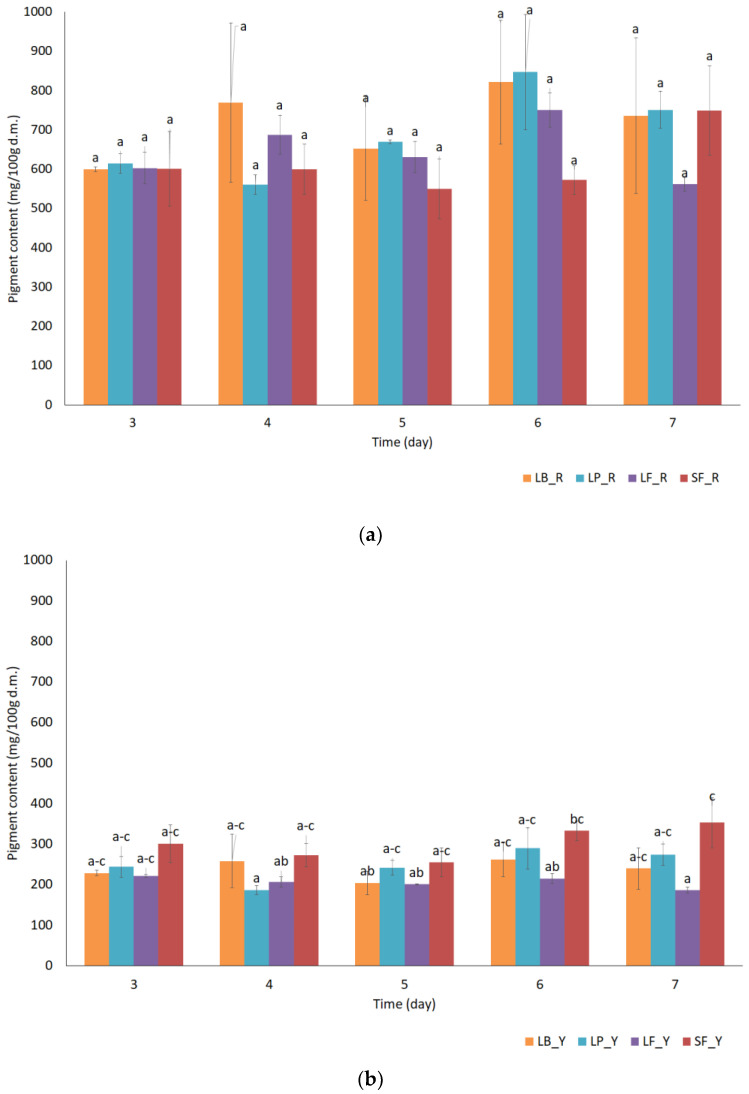
Content of betalain pigments during 7 days of juice fermentation (**a**) red pigment content (**b**) yellow pigment content, SF—spontaneous fermentation, LB—*Levilactobacillus brevis*, LF—*Limosilactobacillus fermentum*, LP—*Lactiplantibacillus plantarum*, _R, _Y—mean red pigment or yellow pigment, respectively, and a, b, c, and other letters—different indexes for pigment mean statistically significant differences for given values at the level of *p* < 0.05.

**Figure 2 molecules-27-01008-f002:**
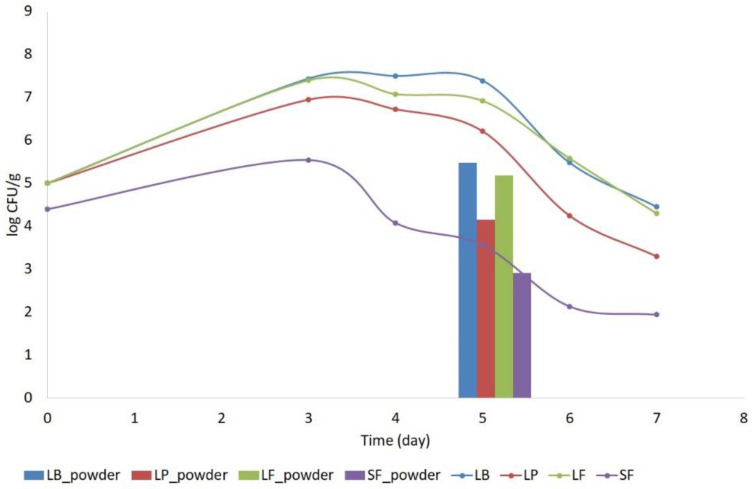
Content of lactic acid bacteria during 7 days of juice fermentation and in spray-dried powders. SF—spontaneous fermentation, LB—*Levilactobacillus brevis*, LF—*Limosilactobacillus fermentum*, LP—*Lactiplantibacillus plantarum*.

**Figure 3 molecules-27-01008-f003:**
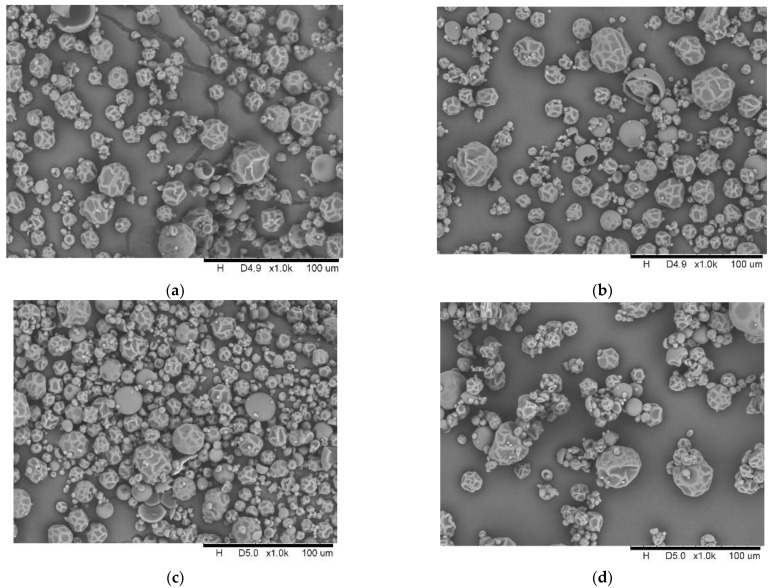
SEM image of powders (**a**) LB_powder; (**b**) LF_powder. (**c**) LP_powder; (**d**) SF_powder; (SF—spontaneous fermentation, LB—*Levilactobacillus brevis*, LF—*Limosilactobacillus fermentum*, LP—*Lactiplantibacillus plantarum*).

**Figure 4 molecules-27-01008-f004:**
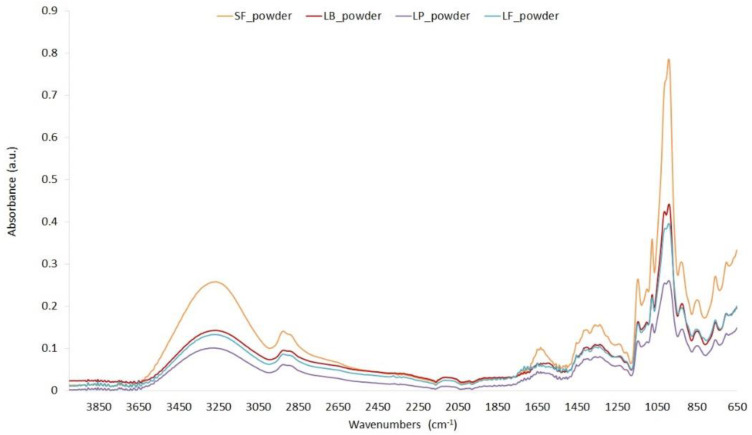
FTIR spectrum of spray dried fermented beetroot powders in the region of 4000–650 cm^−1^.

**Table 1 molecules-27-01008-t001:** Physicochemical properties of fermented beetroot juices.

Sample Name	Extract (° Brix)	pH (-)	Dry Matter d.m. (%)	Total Acidity (g Lactic Acid /100 g Product)	Color Coefficients		
					L*	a*	b*
B_0	7.35 ± 0.07 ^cd^	5.8 ± 0.05 ^g^	4.3 ± 0.1 ^a^	0.35 ± 0.04 ^ab^	5.27 ± 0.04 ^i^	7.37 ± 0.16 ^h^	0.17 ± 0.01 ^bd^
LB_3	7.60 ± 0.00 ^ef^	4.2 ± 0.05 ^e^	5.9 ± 0.1 ^a^	1.09 ± 0.30 ^ef^	4.09 ± 0.04 ^h^	4.88 ± 0.01 ^d^	0.45 ± 0.12 ^eg^
LB_4	6.95 ± 0.07 ^a^	4.1 ± 0.05 ^d^	4.7 ± 1.2 ^a^	0.84 ± 0.01 ^bf^	1.31 ± 0.07 ^b^	6.90 ± 0.07 ^g^	0.07 ± 0.04 ^ac^
LB_5	7.80 ± 0.00 ^g^	4.0 ± 0.05 ^c^	5.7 ± 0.2 ^a^	0.86 ± 0.09 ^cf^	4.02 ± 0.05 ^h^	8.00 ± 0.01 ^i^	0.36 ± 0.08 ^df^
LB_6	7.35 ± 0.07 ^cd^	4.0 ± 0.05 ^c^	4.2 ± 0.7 ^a^	0.72 ± 0.03 ^ae^	3.07 ± 0.08 ^e^	7.01 ± 0.08 ^gh^	0.68 ± 0.10 ^gj^
LB_7	7.70 ± 0.00 ^fg^	3.8 ± 0.05 ^a^	4.3 ± 1.1 ^a^	0.89 ± 0.03 ^cf^	2.74 ± 0.04 ^d^	7.17 ± 0.16 ^gh^	0.51 ± 0.10 ^eh^
LF_3	7.80 ± 0.00 ^g^	4.0 ± 0.05 ^c^	6.1 ± 0.6 ^a^	1.64 ± 0.23 ^g^	4.13 ± 0.04 ^g^	4.07 ± 0.13 ^c^	0.16 ± 0.15 ^bd^
LF_4	7.60 ± 0.00 ^ef^	4.0 ± 0.05 ^c^	5.2 ± 0.1 ^a^	0.86 ± 0.07 ^cf^	1.24 ± 0.05 ^b^	5.61 ± 0.01 ^e^	−0.08 ± 0.08 ^ab^
LF_5	7.65 ± 0.07 ^eg^	3.9 ± 0.05 ^b^	5.1 ± 0.3 ^a^	1.05 ± 0.14 ^df^	4.26 ± 0.04 ^g^	9.79 ± 0.08 ^l^	0.81 ± 0.12 ^ik^
LF_6	7.60 ± 0.00 ^ef^	3.9 ± 0.05 ^b^	4.9 ± 0.2 ^a^	0.80 ± 0.06 ^bf^	3.74 ± 0.06 ^f^	8.81 ± 0.06 ^j^	0.78 ± 0.08 ^hk^
LF_7	7.60 ± 0.00 ^ef^	3.9 ± 0.05 ^b^	5.4 ± 0.2 ^a^	0.70 ± 0.10 ^ae^	3.85 ± 0.08 ^f^	10.47 ± 0.10 ^m^	0.94 ± 0.03 ^jk^
LP_3	7.55 ± 0.07 ^ef^	3.9 ± 0.05 ^b^	5.6 ± 0.1 ^a^	1.59 ± 0.26 ^g^	5.16 ± 0.02 ^i^	9.90 ± 0.13 ^l^	1.38 ± 0.11 ^l^
LP_4	7.30 ± 0.00 ^bc^	3.9 ± 0.05 ^b^	5.6 ± 0.1 ^a^	1.19 ± 0.00 ^eg^	2.12 ± 0.05 ^c^	10.53 ± 0.09 ^m^	0.52 ± 0.05 ^eh^
LP_5	7.70 ± 0.00 ^fg^	3.8 ± 0.05 ^a^	5.0 ± 0.2 ^a^	1.29 ± 0.09 ^fg^	5.74 ± 0.05 ^j^	15.85 ± 0.07 ^o^	1.95 ± 0.05 ^m^
LP_6	7.15 ± 0.07 ^b^	3.9 ± 0.05 ^b^	4.0 ± 0.6 ^a^	1.19 ± 0.16 ^eg^	3.71 ± 0.03 ^f^	9.39 ± 0.08 ^k^	0.95 ± 0.04 ^k^
LP_7	7.50 ± 0.00 ^de^	3.8 ± 0.05 ^a^	4.3 ± 0.2 ^a^	1.27 ± 0.09 ^fg^	3.75 ± 0.07 ^f^	11.04 ± 0.23 ^n^	1.04 ± 0.06 ^k^
SF_3	7.70 ± 0.10 ^fg^	5.9 ± 0.05 ^h^	5.2 ± 0.7 ^a^	0.56 ± 0.06 ^ad^	3.75 ± 0.02 ^f^	2.11 ± 0.07 ^a^	−0.16 ± 0.06 ^a^
SF_4	8.10 ± 0.00 ^h^	5.9 ± 0.05 ^h^	5.3 ± 0.5 ^a^	0.41 ± 0.07 ^ac^	0.81 ± 0.03 ^a^	3.14 ± 0.07 ^b^	−0.04 ± 0.09 ^ab^
SF_5	8.00 ± 0.00 ^h^	5.9 ± 0.05 ^h^	5.7 ± 0.8 ^a^	0.27 ± 0.03 ^a^	3.18 ± 0.02 ^e^	3.44 ± 0.13 ^b^	−0.02 ± 0.07 ^ab^
SF_6	7.70 ± 0.00 ^fg^	5.0 ± 0.05 ^f^	4.6 ± 0.2 ^a^	0.35 ± 0.03 ^ab^	3.10 ± 0.03 ^e^	5.07 ± 0.13 ^d^	0.58 ± 0.00 ^ei^
SF_7	8.00 ± 0.00 ^h^	5.1 ± 0.05 ^f^	4.3 ± 0.6 ^a^	0.24 ± 0.02 ^a^	2.86 ± 0.11 ^d^	6.24 ± 0.30 ^f^	0.31 ± 0.16 ^ce^

* B—beetroot juice on day 0; SF—spontaneous fermentation, LB—*Levilactobacillus brevis*, LF—*Limosilactobacillus fermentum*, LP—*Lactiplantibacillus plantarum*, _0,_3,_4,_5,_6,_7—mean day 0, 3rd day, 4th day 5th day, 6th day, and 7th day of the process, respectively; ^a–c^, and specific letters—different indexes for column mean statistically significant differences for given values at the level of *p* < 0.05.

**Table 2 molecules-27-01008-t002:** Physicochemical properties of fermented beetroot powders.

Sample Name	Dry Matter (%)	Water Activity (-)	Mass Increment after 24 h (%)	Total Acidity (g Lactic Acid /100 g Product)	Lightness L*	Redness a*	Yellowness b*
LB_powder	94.5 ± 0.5 ^b^	0.35 ± 0.01 ^a^	20.8 ± 1.0 ^b^	2.99 ± 0.12 ^a^	43.04 ± 0.01 ^a^	39.14 ± 0.05 ^b^	−18.45 ± 0.00 ^a^
LF_powder	94.2 ± 0.3 ^b^	0.37 ± 0.01 ^a^	16.6 ± 0.5 ^a^	2.85 ± 0.26 ^a^	51.29 ± 0.26 ^c^	34.69 ± 0.25 ^a^	2.64 ± 0.02 ^c^
LP_powder	94.6 ± 0.1 ^c^	0.35 ± 0.00 ^a^	16.5 ± 0.7 ^a^	2.86 ± 0.09 ^a^	48.97 ± 0.06 ^b^	39.18 ± 0.20 ^b^	−18.47 ± 0.12 ^a^
SF_powder	90.2 ± 0.1 ^a^	0.52 ± 0.01 ^b^	19.5 ± 0.8 ^b^	2.96 ± 0.39 ^a^	49.22 ± 0.01 ^b^	38.86 ± 0.01 ^b^	−15.30 ± 0.01 ^b^

SF—spontaneous fermentation, LB—*Levilactobacillus brevis*, LF—*Limosilactobacillus fermentum*, LP—*Lactiplantibacillus plantarum*; ^a–c^, and other letters—different indexes for column mean statistically significant differences for given values at the level of *p* < 0.05.

**Table 3 molecules-27-01008-t003:** HPLC analysis results.

	Betaxanthin mg/100 g d.m.					Sum	Betacyanin mg/100 g d.m.									Sum
	A	B	C	D	E		F	G	H	I	J	K	L	M	N	
Fresh beetroot juice	183.7 ± 1.1	4.8 ± 0.1	6.0 ± 0.1	6.1 ± 0.2	3.5 ± 0.1	204.2 ± 1.4 ^d^	2.0 ± 0.0	1639.0 ± 30.2	3.0 ± 0.2	104.3 ± 0.0	1.2 ± 0.1	0	5.0 ± 0.1	3.7 ± 0.2	0	1758.3 ± 29.8 ^d^
SF_5	1.4 ± 0.1	0	0	0	0	1.4 ± 0.1 ^c^	0	2.0 ± 0.1	0	0.3 ± 0.0	8.4 ± 0.3	0	0	0	0	10.8 ± 0.4 ^a^
LB_5	0.3 ± 0.0	0	0	0	0	0.3 ± 0.0 ^b^	1.6 ± 0.0	79.8 ± 1.6	1.6 ± 0.5	27.1 ± 0.6	1.7 ± 0.1	0	0	1.0 ± 0.1	0	112.7 ± 2.2 ^b^
LP_5	0.1 ± 0.0	0	0	0	0	0.1 ± 0.0 ^a^	1.2 ± 0.0	69.4 ± 4.9	1.1 ± 0.1	26.0 ± 1.1	0.6 ± 0.0	0	0	0.9 ± 0.1	0	99.3 ± 6.0 ^c^
LF_5	0.1 ± 0.0	0	0	0	0	0.1 ± 0.0 ^a^	1.1 ± 0.1	65.3 ± 4.8	1.2 ± 0.0	23.5 ± 1.5	1.0 ± 0.1	0	0	0.8 ± 0.0	0	92.8 ± 6.5 ^c^
SF_powder	2.3 ± 0.1	0	0	0	0	2.3 ± 0.1 ^B^	9.4 ± 0.1	335.2 ± 1.0	6.6 ± 0.0	98.1 ± 2.2	0.9 ± 0.1	1.4 ± 0.1	1.1 ± 0.2	8.2 ± 0.3	13.7 ± 0.3	474.6 ± 1.9 ^A^
LB_powder	0.3 ± 0.0	0	0	0	0	0.3 ± 0.0 ^A^	7.5 ± 0.2	394.5 ± 1.7	6.3 ± 0.3	135.5 ± 0.4	5.4 ± 0.5	1.7 ± 0.1	1.8 ± 0.1	6.3 ± 0.2	1.7 ± 0.1	560.5 ± 1.7 ^C^
LP_powder	0.3 ± 0.0	0	0	0	0	0.3 ± 0.0 ^A^	6.9 ± 0.2	408.4 ± 1.6	7.0 ± 0.3	156.3 ± 0.6	3.2 ± 0.5	0.7 ± 0.1	1.4 ± 0.1	7.2 ± 0.1	1.8 ± 0.0	592.9 ± 1.0 ^D^
LF_powder	0.3 ± 0.0	0	0	0	0	0.3 ± 0.0 ^A^	5.5 ± 0.2	375.0 ± 0.9	6.4 ± 0.0	139.3 ± 2.9	6.3 ± 0.2	1.8 ± 0.0	2.1 ± 0.0	6.0 ± 0.1	2.2 ± 0.0	544.5 ± 4.0 ^B^

SF—spontaneous fermentation, LB—*Levilactobacillus brevis*, LF—*Limosilactobacillus fermentum*, LP—*Lactiplantibacillus plantarum*, _5—fifth day of fermentation, ^a–d^—different indexes for sum column only for juices mean statistically significant differences for given values at the level of *p* < 0.05; ^A–D^—different indexes for sum column only for powders mean statistically significant differences for given values at the level of *p* < 0.05. A—vulgaxanthin I, B—other betaxanthin, C—other betaxanthin, D—other betaxanthin, E—other betaxanthin, F—possible decarboxylated derivative of betanin, G—betanin, H—other betacyanins, I—isobetanin, J—betanidin, K—other betacyanins, L—other betacyanins, M—neobetanin, and N—other betacyanins.

**Table 4 molecules-27-01008-t004:** TGA—thermal decomposition of powders.

Sample Name	Step 1		Step 2		Step 3		Sum [%]	Decomposition Temperature (°C)			
	Temp. Range (°C)	Mass Loss (%)	Temp. Range (°C)	Mass Loss (%)	Temp. Range (°C)	Mass Loss (%)		1	2		
LB_powder	30–120	0	120–420	56.7	420–600	1.6	58.3	63.8	160.2	202.9	286.5
LF_powder	30–120	2.9	120–420	57.5	420–600	1.3	61.7	62.9	160.5	207.2	287.6
LP_powder	30–120	3.5	120–420	56.2	420–600	0.9	60.6	60.4	160.5	206.5	289.2
SF_powder	30–120	4.9	120–420	52.5	420–600	5.7	63.1	42.7	167.5	-	272.5

LB—Levilactobacillus brevis, LF—Limosilactobacillus fermentum, LP—Lactiplantibacillus plantarum, SF—spontaneous fermentation.

## Data Availability

Data available from the corresponding author.

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
