# Peer review of "Influence of Fermentation Beetroot Juice Process on the Physico-Chemical Properties of Spray Dried Powder"

_molecules, 2022, doi:10.3390/molecules27031008_

Round 1

Reviewer 1 Report

The manuscript submitted by Janiszewska-Turak and collaborators present a work regarding influence of the fermentation process on the colorants amount in fermented beetroot juice and its spray dried powder.

There are several major questions from the reviewer as below:

  1. Avoid the use of abbreviations without prior identification.
  2. The introduction does not match with the title, I suggest the authors to modify the introduction focusing on the state-of-the-art of fermentation processes to produce colorants from beetroot.
  3. The main objective of the present work is not clear.
  4. Line 111: Avoid the use of abbreviations without prior identification
  5. In the caption of table 2, what is the meaning of 4th ect?
  6. In the same caption, the authors declare that a, b – different indexes for column mean statistically significant, what about c,d,e,f,g,h………
  7. The authors should discuss more the results obtained from LF, LP, LB, SF separately, and then discuss the main differences between them.
  8. The results from fig 1 are confused, I suggest the authors to separate the results in order to avoid misunderstanding.
  9. Based on the results from Fig 2, the stationary phase start in day 4, why the author select day 5?
  10. In the section 2.2, why the author selects day 5th over the 4th?
  11. Where the data of decomposition temperature of SF? The authors don’t explain why they use TGA and other procedures.
  12. I reiterate that the discussion of this article should be greatly improved, viz., explaining the impact of differences in morphology, parameters that influence the production of colorants, influence of these parameters on the production of each biomolecule, etc.….. and where’s the innovation of this work.

To sum up, in my opinion the design of this experiments is rational, but the discussion of the results is not enough to include all the possible factors during the process. Furthermore, for the guidance of future manufactory studies, the claims in this paper should be more convincing. The discussion of this study is still weak in the present form, which failed to meet the requirement of the journal Molecules.

Author Response

 Dear reviewer

We thank you for the review and for all comments and suggestions. Answers are listed below. The whole manuscript was checked by professional English interpreter.

Reviewer 2 Report

  • The title is not appropriate, because “fermentation process” is only to study different strain of lactic acid bacteria. The fermentation time was selected 5th day for four different fermented beetroot juice.
  • The colorant amount such as betalain and vulgaxanthin in fermented beetroot juice and spray dried power did not show the individual value, and only represented by pigment loss by fluorescence.
  • The pigment content change significantly during the spray drying processing due to 10% maltodextrin addition and decreasing water evaporation; however, there was no significant difference among these spray dried powders.
  • If the pictures of original, four fermented beetroot juice and spray dried powder  can be provided for comparing the color difference will be better.
  • It is difficult to control and analyze the lactic acid bacteria strain in spontaneous fermentation to lead different pH, acid and color  comparing other fermented product. Therefore, it can be deleted and easily discussion.
  • It is clear to discuss the changes of spray dried sample. 

Author Response

(The authors gave the same response as above.)

Round 2

Reviewer 1 Report

There was a big improvement in the quality of the manuscript. However, the authors did not consider all my comments, viz., explaining the influence of the production parameters on the production of each biomolecule, etc...

Furthermore, the authors revealed that the day five was chosen based on data from earlier preliminary studies. I recommend the addition of the data in supplementary material for better understanding of readers.

Figure 3 can be removed from the main manuscript. I recommend to send to the supplementary material.

Author Response

There was a big improvement in the quality of the manuscript. However, the authors did not consider all my comments, viz., explaining the influence of the production parameters on the production of each biomolecule, etc...

Dear reviewer

We thank you for the review and for all comments and suggestions. We have tried to answer the best we could at each previous question. We apologise if answers did not satisfied you.
We have added now more information about Betalain (in our article this is biomolecule) and its changes during fermentation and spray drying.

Furthermore, the authors revealed that the day five was chosen based on data from earlier preliminary studies. I recommend the addition of the data in supplementary material for better understanding of readers.

The data for betalain content and bacteria count has been placed in supplementary file, as well as the explanation

Figure 3 can be removed from the main manuscript. I recommend to send to the supplementary material

It is now in supplementary file.

Reviewer 2 Report

I accept the revised manuscript. The title of manuscript was well changed to "Influence of fermentation beetroot juice process on the physico-chemical properties of spray dried powder" by authors.

Author Response

Thank you very much for all comments which have improved the manuscript.